# In Silico Study Probes Potential Inhibitors of Human Dihydrofolate Reductase for Cancer Therapeutics

**DOI:** 10.3390/jcm8020233

**Published:** 2019-02-11

**Authors:** Rabia Mukhtar Rana, Shailima Rampogu, Amir Zeb, Minky Son, Chanin Park, Gihwan Lee, Sanghwa Yoon, Ayoung Baek, Sarvanan Parameswaran, Seok Ju Park, Keun Woo Lee

**Affiliations:** 1Division of Life Sciences, Division of Applied Life Science (BK21 Plus), Research Institute of Natural Science (RINS), Gyeongsang National University (GNU), 501 Jinju-daero, Jinju 52828, Korea; rabia.mukhtar.rana@gmail.com (R.M.R.); shailima.rampogu@gmail.com (S.R.); zebamir85@gmail.com (A.Z.); minky@gnu.ac.kr (M.S.); chaninpark0806@gmail.com (C.P.); pika890131@gmail.com (G.L.); jsyoon0517@gmail.com (S.Y.); ayoung@gnu.ac.kr (A.B.); dr.p.saravanan.bi@gmail.com (S.P.); 2Department of Internal Medicine, College of Medicine, Busan Paik Hospital, Inje University, Busan 47392, Korea

**Keywords:** dihydrofolate reductase inhibition, pharmacophore modeling, molecular docking, molecular dynamics simulation, binding free energy

## Abstract

Dihydrofolate reductase (DHFR) is an essential cellular enzyme and thereby catalyzes the reduction of dihydrofolate to tetrahydrofolate (THF). In cancer medication, inhibition of human DHFR (hDHFR) remains a promising strategy, as it depletes THF and slows DNA synthesis and cell proliferation. In the current study, ligand-based pharmacophore modeling identified and evaluated the critical chemical features of hDHFR inhibitors. A pharmacophore model (Hypo1) was generated from known inhibitors of DHFR with a correlation coefficient (0.94), root mean square (RMS) deviation (0.99), and total cost value (125.28). Hypo1 was comprised of four chemical features, including two hydrogen bond donors (HDB), one hydrogen bond acceptor (HBA), and one hydrophobic (HYP). Hypo1 was validated using Fischer’s randomization, test set, and decoy set validations, employed as a 3D query in a virtual screening at Maybridge, Chembridge, Asinex, National Cancer Institute (NCI), and Zinc databases. Hypo1-retrieved compounds were filtered by an absorption, distribution, metabolism, excretion, and toxicity (ADMET) assessment test and Lipinski’s rule of five, where the drug-like hit compounds were identified. The hit compounds were docked in the active site of hDHFR and compounds with Goldfitness score was greater than 44.67 (docking score for the reference compound), clustering analysis, and hydrogen bond interactions were identified. Furthermore, molecular dynamics (MD) simulation identified three compounds as the best inhibitors of hDHFR with the lowest root mean square deviation (1.2 Å to 1.8 Å), hydrogen bond interactions with hDHFR, and low binding free energy (−127 kJ/mol to −178 kJ/mol). Finally, the toxicity prediction by computer (TOPKAT) affirmed the safety of the novel inhibitors of hDHFR in human body. Overall, we recommend novel hit compounds of hDHFR for cancer and rheumatoid arthritis chemotherapeutics.

## 1. Introduction

Dihydrofolate reductase (DHFR) is a ubiquitous enzyme and exists in a wide range of organisms [1]. DHFR is crucial for proper cellular growth and proliferation, where it regulates the maintenance of tetrahydrofolate (THF) and its derivatives, leading to the synthesis of purine and thymidylates [2]. DHFR is known for its enzymatic action to catalyze the reduction of 7,8-dihydrofolate (DHF) to 5,6,7,8-THF. The reaction process is divided into two steps. First, protonation of DHF is taking place at N(5) atom of the pteridine moiety. In second step, a hydride ion (2H^−^) transfers from the coenzyme nicotenamide adenine dinucleotide phosphate hydrogen (NADPH) to the C6 atom of the pteredine ring of the DHF [3,4]. DHFR is also responsible for the reduction of folate to DHF, but the efficiency is very low [5]. During the folate reduction by DHFR, the protonation of pteridine ring is occurring at position N(8). 

The expression of human DHFR gene synthesizes DHFR enzyme and is located on q22 of chromosome 5. Human DHFR (hDHFR) has a high sequence similarity with mammalian DHFR (~70%) but a low similarity with bacterial DHFRs (~30%) [6]. hDHFR is comprised of eight β sheets and four α helices. Among the four α helices, two helices make the substrate binding site, while the remaining two helices form the binding domain of the coenzyme NADPH. The active site of hDHFR is made up of alpha helix B, the central beta sheet, and the loop 1. In the folate binding site of hDHFR, DHF is bound with an almost planar arrangement of its p-aminobenzamide moiety. This is the preferred conformation of this moiety in crystalline folic acid and enzyme bound folic acid derivatives [7]. DHF occupies an extended cavity located on one face of the central sheet. Its surrounding is dominated by hydrophobic residues of the α-helical segments α1 (Arg28 to Thr39) and α11 (Lys54 to Ser59), as well as the irregular part between the Ser59 and Ile71 and the central β strands. Hydrophobic contacts are formed with the bulky side chains of Phe31 and Phe34, which cover one face of the pteridine ring [8]. Amino acids involved in the most relevant molecular interactions of substrate and/or inhibitor(s) binding are Ile7, Glu30, Phe31, Phe34, Leu67, Arg70, and Val115.

Despite the extensive targeting of DHFR in pathogenic organisms [9,10,11,12], hDHFR is also targeted by anticancer agents [2,13,14,15]. Methotrexate (MTX) targets hDHFR and is a well-known anticancer drug for the treatment of leukemia, breast cancer, lung cancer, osteosarcoma, and lymphoma [16]. MTX is a strong competitor of DHF with regard to binding hDHFR, as it causes elevated accumulation of DHF when it binds hDHFR, which in turn induces the feedback inhibition of thymidylate synthase. Singh et al. modified an antibacterial agent trimethoprim to compounds 2 and 3 with promising anticancer applications [17]. In another attempt, the quinazoline analogs were synthesized to have closed resemblance with MTX and inhibited the mammalian DHFR in cancerous cells [18]. The application of these quinazoline analogs was also recommended for combined inhibition of epidermal growth factor receptor (EGFR) and hDHFR. In spite of anticancer therapy, hDHFR is also known to be an important target for the treatment of rheumatoid arthritis [19,20]. 

Nevertheless, MTX is a promising hDHFR inhibitor, but it has shown several side effects including line fever, nausea, fatigue, skin infection, and low white blood cell (WBC) count [21]. The long term use of MTX causes severe medical consequences such as kidney, skin rashes, and liver and lung diseases. Furthermore, it is not safe for ladies with breast feeding [22,23,24]. Because of the severe health issues as side effects, decrease in efficiency, and the failure of its analogues in successful anti-DHFR agents, the discovery of new and safe drug-like candidate molecules of hDHFR is needed. To this end, we used computer aided drug designing approaches to identify novel and potent candidate molecules that can potentially inhibit hDHFR. A 3D-QSAR pharmacophore model (Hypo1) was built from already known inhibitors of the hDHFR [25]. Hypo1 was validated and used as a three-dimensional (3D) query in virtual screening of the chemical databases. The Hypo1-retrieved chemical compounds from Maybridge, Chembridge, Asinex, NCI, and Zinc databases, which were subjected to Lipinski’s rule of five and the absorption, distribution, metabolism, excretion, and (ADMET) descriptors modules of the Discovery Studio v.4.5 (Dassault System, BIOVIA Corp, San Diego, CA, USA) and the drug-like hit molecules of DHFR were identified. The binding affinity and conformation of the selected hit compounds in the active site of hDHFR were predicted by molecular docking studies. Finally, three hit molecules of the hDHFR were identified using a molecular dynamic (MD) simulation. The simulation of each candidate hit hDHFR molecules obtained from a stable root mean square deviation (RMSD). Each hit molecule established strong hydrogen bond(s) with the catalytic active residues of the hDHFR. The binding free energy calculations suggested that the final hit molecules strongly bind hDHFR. Overall, we recommend that the newly identified candidate hit molecules of hDHFR may serve as the fundamental approaches to design more efficient and safe inhibitors of the hDHFR. Such inhibitors may have potential applications in cancer chemotherapy. 

## 2. Materials and Methods

### 2.1. Collection of Dataset

A dataset of 67 known inhibitors of hDHFR was accumulated from literature mining (https://www.bindingdb.org/bind/index.jsp) and categorized into two distinctive data sets: (i) a training set and (ii) a test set. The training set compounds were sub-categorized into active (IC_50_ < 100 nM/L), moderate active (100 nM/L < IC_50_ < 500 nM/L), and inactive (IC_50_ ≥ 500 nM/L) compounds on the basis of structural diversity and experimentally determined IC_50_ values. The training set compounds were used for pharmacophore generation. The test set compounds were used to validate the hypotheses. The 2D structures of all the compounds were drawn by Biovia Draw and subsequently converted to their 3D structures in Discovery Studio v4.5 (DS).

### 2.2. Generation of Pharmacophore Model

In this study, the 3D QSAR Pharmacophore Generation module of DS was used to generate ligand-based pharmacophoric hypotheses. In brief, the possible pharmacophoric features of the training set compounds were identified with Feature Mapping protocol of DS. The training set was used for hypotheses generation through the HypoGen algorithm of 3D QSAR Pharmacophore Generation module, which was implanted in DS. During the hypotheses generation, the minimum and maximum numbers of features were set to 0 and 5, respectively. BEST algorithm of DS was employed to generate low energy conformations of the training set compounds. Uncertainty value was set to 3 while other parameters had default values. Hypotheses were generated with their resultant statistical parameters such as cost values (fixed cost and null cost), correlation (*R*^2^), root mean square deviation (RMSD), and fit values. Cost values were analyzed as per Debnath’s method [26].

### 2.3. Pharmacophore Validation

Hypothesis validation is carrying out to evaluate the quality of pharmacophore in computational drug designing procedures. The top-ranked hypothesis was validated by Fisher’s randomization, test set method, and decoy test method. The statistical significance of the model was computed by employing Fischer’s randomization method. The confidence level was set to 95% and 19 random spreadsheets were generated by arbitrarily reassigning the experimental activity values to each compound in the training set. The significance of hypothesis is calculated as:S = [1 − (1 + X)/Y] × 100(1)
where X denotes the total number of hypotheses with a total cost lower than the original hypothesis and Y represents the total number of HypoGen runs (initial + random runs). Here, X = 0 and Y = (1 + 19), hence 95% = {1 − [(1 + 0)/(19 + 1)]} × 100.

In test set validation, the selected pharmacophore hypothesis was used to predict the activity values of the test set compounds. The predicted and experimental activity values were plotted to observe the range of correlation between them. Additionally, the decoy set method was recruited to further affirm the robustness of the model and was estimated as:EF = (Ha/Ht)/(A/D)(2)
GF = (Ha/4HtA) (3A + Ht) × [1 − (Ht − Ha)/(D − A)](3)
where D represents the total number of molecules in the data set, A indicates the total number of active molecules (hDHFR inhibitors) in the data set, Ht denotes the total number of hits retrieved, and Ha refers to the number of active molecules present in the retrieved hits.

Furthermore, the efficacy of the Hypo1 was determined by the enrichment factor (EF) score and the goodness of fit (GF) score.

### 2.4. Virtual Screening and Drug-Likeness Prediction

The validated hypothesis was used as a 3D structural query in the virtual screening of chemical databases, including Chembridge, NCI, Asinex, Maybridge, and ZINC database. For virtual screening, the Ligand Pharmacophore Mapping tool of DS was employed. During the screening, Conformation Generation and Fitting Methods were kept to BEST and Flexible modes, respectively. The Maximum Omitted Feature was set to 0.

Chemical compounds should have acceptable physiochemical, pharmacokinetic, and pharmacodynamics properties to be used as drug molecules. The drug-like properties of the screened compounds were evaluated with ADMET Descriptors and Lipinski’s rule of five modules of DS. During the ADMET filtration, estimated values of blood-brain barrier (BBB), optimal solubility, and absorption levels were set to 3, 3, and 0, respectively. Other parameters such as CYP2D6 prediction and hepatotoxicity prediction were assigned as false. In Lipinski’s rule of five, the protocol was optimized as the number of hydrogen bond donors was less than 5, number of hydrogen bond acceptor was less than 10, molecular weight was equal or less than 500 Da, and AlogP was selected to equal or less than 5. The successful molecules were assigned as the hit compounds and subjected to further analyses. 

### 2.5. Molecular Docking Simulation

Docking studies were performed using Genetic Optimization of Ligand Docking (GOLD) package v5.2.2 (The Cambridge Crystallographic Data Centre) [27]. For molecular docking, the crystal structure of hDHFR in complex with inhibitor N6-(2,5-Dimethoxy-Benzyl)-N6-Methyl-Pyrido[2,3-D]Pyrimidine-2,4,6-Triamine (PRD) was taken from protein data bank (PDB ID: 1BOZ). During the selection of hDHFR structure, it was confirmed that when the structure was completed, no mutations were observed, having high resolution and the bound inhibitor. The structure of hDHFR was prepared for docking by removing water molecules and hetero atoms in DS. Chemistry at Harvard macromolecular mechanisms (CHARMm) force field was used to add hydrogen atoms to the structure of hDHFR. The binding site of hDHFR was identified from the bound inhibitor (PRD) using the Define and Edit Binding Site module, seeded in DS. During docking, the hit compounds along with the training set compounds were treated as ligand molecules. The Goldfitness score and Chemscore functions of GOLD were assigned as the default scoring and rescoring functions, respectively. For each ligand, a total of 250 conformers were generated using Genetic Algorithm (GA) of the GOLD package. Docking results were analyzed with DS.

### 2.6. Molecular Dynamics (MD) Simulation

MD simulations were carried out with a CHARMm27 all-atom force field [28] in Groningen Machine for Chemical Simulation (GROMACS) v5.0.6 package [29]. For each ligand, an independent simulation system was generated. The topology and coordinates files for the co-factor (NADPH) of hDHFR and hit molecules were generated using SwissParam. Each system was solvated with transferable intermolecular potential with three points (TIP3P) water model in a octahedral box. Solvent molecules were replaced with sodium ions (Na^+^) to neutralize the simulation systems. The energy minimization of the simulation systems were carried out using the steepest descent algorithm at a maximum force less than 10 kJ/mol to avoid any possible bad contacts during the production phase. Equilibration of simulation systems was performed in two phases. First, number of particles at constant volume and temperature (NVT) equilibration was conducted for 300 ps at 300 K. A V-rescale thermostat was used to maintain constant temperature [30]. Secondly, number of particles at constant pressure and temperature (NPT) equilibration was performed for 300 ps at constant pressure of 1 bar [31]. Finally, each system was simulated while following the protocol previously described. In brief, the Linear constraint solver (LINCS) algorithm was used to preserve the bond length of heavy atoms. The long range electrostatic interactions were computed by employing the mesh ewald (PME) method. The threshold value of short-range interactions was kept to 14 Å. All simulations were executed under the periodic boundary conditions to make infinite systems. The coordinate data were saved at 1 ps time interval. The results were visualized using GROMACS and DS.

### 2.7. Binding Free Energy Calculations

The Molecular Mechanics/Poisson-Boltzmann Surface Area (MM/PBSA) method was used for binding energy calculations [32]. Equidistant snapshots of hDHFR-ligand complex were extracted and MM/PBSA protocol was followed for binding energy calculations. 

The binding free energy of protein-ligand complex is stated as:Δ*G*_binding_ = *G*_complex_ − (*G*_protein_ + *G*_ligand_),(4)
where, *G*_complex_ refers to the sum of the free energy of the complex, and *G*_protein_ and *G*_ligand_ indicate the free energies of portion and ligand in their unbound state. 

The free energy can be expressed as:*G*_*X*_ = *E*_MM_ + *G*_solvation_,(5)
where *X* can be a protein, ligand, or their complex. *E*_MM_ represents the average molecular mechanics potential energy in vacuum, while *G*_solvation_ interprets the free energy of solvation.

Additionally, molecular mechanics potential energy in vacuum can be evaluated by adapting the equation:*E*_MM_ = *E*_bonded_ + *E*_non-bonded_ = *E*_bonded_ + (*E*_vdw_ + *E*_elec_),(6)

*E*_bonded_ represents the bonded interactions, while *E*_non-bonded_ depicts the non-bonded interactions. The value of Δ*E*_bonded_ is mostly treated as zero.

The combined energetic terms of electrostatic (*G*_polar_) and apolar (*G*_non-polar_) solvation free energies forms the solvation free energy and is measured as: *G*_solvation_ = *G*_polar_ + *G*_non-polar_,(7)

The Poisson-Boltzmann (PB) equation is employed to compute *G*_polar_, while the *G*_non-polar_ is calculated from the solvent accessible surface area (SASA).

Furthermore, the *G*_non-polar_ calculated as:*G*_non-polar_ = *γ*SASA + *b*,(8)
where, *γ* represents the coefficient of solvent surface tension, while *b* is its fitting parameter, whose values are 0.02267 kJ/mol/ Å2 and 3.849 kJ/mol, respectively.

## 3. Results

### 3.1. Generation of Pharmacophore Model

The training set was designed by selecting compounds from the literature mined dataset. The selection was made on the basis of structural diversity and variation in IC_50_ values of the hDHFR inhibitors. The training set was comprised of 27 inhibitors of the hDHFR and their experimental IC_50_ values were determined using the same biological assays. The 2D structures and anti-DHFR activities (IC_50_ values) ranged from 0.19 to 10,000 nM/L of the training set compounds (Figure 1). The hDHFR inhibitors of the training set were sub-divided into active (IC_50_ < 100 nM/L, +++), moderately active (100 nM/L < IC_50_ < 500 nM/L, ++), and inactive (IC_50_ ≥ 500 nM/L, +) compounds. A total number of hypotheses were generated using a 3D QSAR Pharmacophore Generation module of DS, using the training set compounds (Figure 1). Our results observed that the statistical parameters of Hypo1 included configuration cost (17.06), total cost (125.276), close to the fixed cost (109.01), and away from the null cost (203.67). Futher, a high correlation coefficient of 0.94 went along with a large cost difference (78.308) and a low RMS value (0.99) (Table 1).

The comparative analysis suggested that Hypo1 is the best representative hypothesis, presenting a good geometric spatial agreement with the active site of hDHFR and comprised of four chemical features including 1 hydrogen bond acceptor (HBA), 2 hydrogen bond donors (HBD), and 1 hydrophobic (HYP) (Figure 2A). Hypo1 was selected as the representative hypothesis for further analysis. It is assumed that a best hypothesis (pharmacophore) estimates either the same or closed activity scale (pIC_50_ value) to experimentally determine the IC_50_ value of the tested compound; it is characterized as a predictive accuracy. To elucidate the predictive accuracy of Hypo1, we estimated the activity values of the training set compounds. Our results demonstrated that Hypo1 estimated the same range of inhibitory activity for all the training set compounds except for one active and two moderately active compounds, which were underestimated as moderately active and inactive compounds, respectively (Table 2). The fit value shows the strength of mapping of the correspondent compound onto Hypo1 (Table 2). Error measures the difference between the experimentally determined IC_50_ and the predicted IC_50_ values of the tested compound. The activity scale is represented as active (+++), moderately active (++), and inactive (+) (Table 2).

Additionally, mapping of the training set by Hypo1 showed that all the hypothetical features of Hypo1 were perfectly mapped by the most active (IC_50_ = 0.19 nM/L) compound (Figure 2B), whereas the inactive (IC_50_ = 10,000 nM/L) compound failed to fit on one HBA and one HYP features (Figure 2C). This analysis advocates that Hypo1 has high accuracy to effectively differentiate the chemical compounds on the basis of activity values.

### 3.2. Pharmacophore Validation

#### 3.2.1. Fischer’s Randomization Test

Fischer’s test was carried out to estimate the statistical significance of Hypo1. At 95% confidence level, 19 random spreadsheets were estimated. The estimated random spreadsheets indicate total cost values of 10 generated hypotheses (Figure 3). The comparative analysis showed that the total cost value of Hypo1 was significantly lower than the generated random spreadsheets. Therefore, we argue that Hypo1 is far superior and was not generated by chance.

#### 3.2.2. Test Set Validation

Hypo1 was validated after evaluating its ability to measure the activity range of the test set compounds. The test set was comprised of 40 chemically diverse compounds and sub-divided into active, moderately active, and inactive compounds (Appendix A). Our findings observed that Hypo1 estimated the acceptable activity range of the entire test set compounds except for three molecules (Appendix A). Furthermore, a high correlation coefficient (*R*^2^) value of 0.92 was obtained by linear regression between the experimental inhibitory activities and predicted inhibitory activities of the test set compounds (Figure 4). Our results affirmed that Hypo1 has a strong capacity to differentiate active compounds from moderately active and inactive compounds.

#### 3.2.3. Decoy Set Validation

Decoy set validation was performed using a Ligand Pharmacophore Mapping module in DS. The strength of Hypo1 was determined by four parameters such as false positive, false negative, enrichment factor (EF), and goodness of fit (GF). EF and GF were calculated from the values of different parameters given in Table 3. Other characteristics of Hypo1 including percentage of number of active yields (%Y), percent ratio of actives in the hit list (%A), false negatives, and false positives were also measured (Table 3).

Hypo1 retrieved 87% active compounds from the decoy set. Hypo1 predicted 1 active compound as inactive (false negative) and 1 inactive compound as active (false positive) compound. Hypo1 achieved a high GF score of 0.86. Similarly, Hypo1 obtained enrichment factor (EF) score of 8.23 (Table 3). These observations suggested that Hypo1 had high potential to show true positives.

### 3.3. Virtual Screening of Chemical Databases

Chemical features of pharmacophore play an essential role in the screening of chemical databases and identify the best-fitted compounds from a database. Herein, Hypo1 was employed to screen NCI, Maybridge, Chembridge, Asinex, and ZINC databases, which contains 238,819, 59,652, 50,000, 213,262, and 137,077 compounds, respectively (Figure 5). Hypo1 retrieved a total of 32,617 compounds, which were subsequently subjected to the ADMET assessment test and Lipinski’s Rule of five. The ADMET assessment test evaluated the pharmacokinetic properties of Hypo1-retrieved compounds. During the ADMET assessment test, compounds were checked for non-inhibition to CYP2D6 and non-hepatotoxicity. Other pharmacokinetic properties included blood brain barrier (BBB), optimal solubility, and good intestinal absorption, evaluated by setting their values to 3, 3, and 0, respectively. The ADMET assessment test was followed by Lipinski’s rule of five. During Lipinski’s rule of five filtration, compounds with AlogP value less than 5, number of HBD < 5, number of HBA < 10, molecular weight < 500 Da, and fewer than ten rotatable bonds were identified [33,34]. Finally, a total of 479 molecules satisfied the drug-like properties (Figure 5).

### 3.4. Molecular Docking Simulation

To unveil the binding mode of true positive candidate hits of hDHFR, the Hpo1-retrieved drug-like hits (479) were subjected to molecular docking. The 3D structure of hDHFR in complex with inhibitor PRD was taken from protein databank (PDB ID: 1BOZ) [35]. The resolution of the selected structure is 2.1 Å. The docking protocol was optimized by docking the co-crystal bound inhibitor (PRD) in the active site of hDHFR. The docked pose of the PRD obtained an acceptable RMSD value of 0.92 Å with the crystallographic pose of PRD in the active site of the hDHFR (Appendix A). Furthermore, the potentiality of the generated pharmacophore (Hypo1) model was assessed by its comparative analysis with the docked pose of the reference compound (the most active compound of the training set). Herein, the superimposition of Hypo1 and the docked pose of the reference compound observed an acceptable RMSD value of 1.16 Å (Appendix A). Therefore, the Hypo1-retrieved drug-like (candidate) compounds were docked while employing the same optimized protocol. The docking results showed that the Goldfitness score and Chemscore of the reference compound in the training set were 44.67 and −23.35, respectively, and used as cut-off values during the docking results analysis (Table 4).

The candidate compounds were selected based on a Goldfitness score greater than 44.67, Chemscore lower than −23.35, and the ligand conformations satisfying the necessary interactions with essential residues in the active site of the hDHFR. During virtual screening, hit molecules with estimated IC_50_ values less than the estimated IC_50_ value of the highly active compound were considered. Our results also showed that the selected hit compounds had a lower estimated IC_50_ values than the estimated IC_50_ value (0.19 nM/L) of the highly active compound of the training set (Table 4). Finally, three compounds were selected as the best hit compounds against the hDHFR, which also mapped well onto the pharmacophoric features of Hypo1 (Figure 6).

### 3.5. Molecular Dynamics Simulation

MD simulations were performed to evaluate the binding stability of the novel hits in the active site of the hDHFR. A total of four MD simulation systems were prepared as one for each hit compound and one for the reference compound. The preliminary details of each system are provided in Table 5. 

The stability of simulation system was assessed by measuring the root mean square deviation (RMSD) of the backbone atoms of the hDHFR. The simulation results observed that the root mean square deviation values of backbone atoms of hDHFR ranged from 1.2 Å to 1.8 Å throughout the simulation; each system was well converged (Figure 7A). For further analyses, the representative structure of each system was taken from the last 5 ns trajectory. The superimposition of representative structures observed that the binding pattern and conformational orientation of hit compounds in the active site of hDHFR were similar to the reference compound (Figure 8). The substrate binding site of DHFR is comprised of Ile7, Glu30, Phe34, Ser59, Pro61, Asn64, and Val115 [36]. Our results suggested that the substrate binding residues of hDHFR could bind the reference compound and the hit molecules. The reference compound formed hydrogen bonds with Ile7, Glu30, and Val115 of hDHFR (Figure 9A, Table 6).

Furthermore, the reference compound established π-cation interactions with Ile7, Phe34, Ile60, Pro61, Leu67, and Val115 and showed van der Waals interactions with Val8, Gly31, Asn64, Gly116, Tyr121, and NADPH (Table 6). Hit1 formed hydrogen bond interactions with Leu27, Glu30, Ser59 and van der Waals interaction with Val8, Ile16, Gly17, Asp21, Asp21, Pro26, Gly31, Gln35, and Thr136 (Figure 9B, Table 6). Our findings also suggested that Hit1 formed carbon-hydrogen bonds with Pro61, Ile60, and π-cation interactions with Ala9, Leu22, Pro23, Phe34, Val115, and NADPH. Hit2 formed hydrogen bonds with Trp24, Glu30, Thr56, and NADPH as well as carbon-hydrogen bonds with Gly31, Ser59, Ile60, and Pro61 (Figure 9C, Table 6). Additionally, Hit2 showed interactions with the hydrophobic pocket residues of hDHFR such as Ala9 and Gln351, as well as π-π interactions with Phe34, Met52, Ile60, Leu67, and Val115 (Table 6). Hit3 established hydrogen bonds with Glu30 and Asn64 (Figure 9D, Table 6). Hit3 showed hydrophobic interactions with Val259, Ala271, Glu288, Met292, Ile314, Phe318, Asn369, Leu371, Ala381, and Phe383 residues of the hDHFR. The benzene ring of Hit3 participated in π-cation interactions with Ile7, Ala9, Ile60, and Phe34. Hit3 exhibited van der Waals interactions with the residues forming the hydrophobic core of hDHFR and NADPH, while one carbon-hydrogen bond with Glu30 (Table 6). During the entire simulation period, the total number of intermolecular hydrogen bonds of hDHFR with each hit compound were measured. Our results observed that the hit compounds formed comparatively high number of H-bonds with hDHFR than the reference compound (Figure 7B). To assist the novelty of hit compounds against the hDHFR, PubChem structure was used [34]. Our search confirmed that the newly identified hit compounds have not been previously tested to inhibit hDHFR. Therefore, we recommend these hit compounds as potential candidate inhibitors of hDHFR. The 2D structures of hit compounds are shown in Figure 10.

### 3.6. Binding Free Energy Analysis

The binding free energy of hDHFR in complex with hit molecules ranged from −127.05 kJ/mol to −178.47 kJ/mol (Figure 11A). Our findings observed that the average binding free energy for hDHFR in complexes with reference compound, Hit1, Hit2, and Hit3 were −127.05 kJ/mol, −178.47 kJ/mol, −171.12 kJ/mol, and −133.16 kJ/mol, respectively. The decomposition analysis of the binding free energy suggested that electrostatic and van der Waals forces are pre-dominant factors in DHFR inhibition (Figure 11B, Table 7).

Overall, our results demonstrated that the newly identified hits preferentially accommodated the active site of hDHFR and established polar as well as non-polar interactions with the catalytic active residues.

### 3.7. Toxicity Evaluation by TOPKAT

Toxicity prediction by komputer assisted technology (TOPKAT) is a toxicity prediction tool implanted in DS. TOPKAT is used by universities, private companies, and government agencies such as Amgen, Pfizer, and US CDC to evaluate the toxicity of target molecules. Our results suggested that the newly identified hit compounds of hDHFR are non-carcinogenic and do not pose any skin irritancy. The safety of novel hits of hDHFR was evaluated using several parameters in Table 8.

## 4. Discussion

Inhibitors of folate metabolism are widely recognized as effective drugs of cancer, rheumatoid arthritis, and microbial infections. Though, several enzymes such as thymidilate synthase, DHFR, and serine hydroxymethyl transferase are therapeutic targets of folate metabolism-associated diseases. Nonetheless, DHFR is an essential key enzyme in the DNA and amino acid metabolism [37]. 

In current study, potential inhibitors of hDHFR were identified by 3D-QSAR pharmacophore modeling. 3D-QSAR pharmacophore modeling has become an increasingly acceptable approach for probing novel lead candidates in computational drug designing [38]. A pharmacophore model is considered to be reliable if it has low RMS values, high cost difference, good correlation coefficient, the lowest total cost, and if the total cost is close to the fixed cost and far from the null cost [39,40]. Accordingly, Hypo1 was selected as the best hypothesis, which was comprised of four chemical features including HBA, HBD, and HYP. Pharmacophore validation is an essential step in computational drug designing [41]. Fischer’s randomization method, test set validation, and decoy test methods are state-of-the-art methods to validate the quality of the pharmcophore. The validation of pharmacophore obtained the similar results as previously described in computational drug designing [35,42]. In brief, the high correlation coefficient (0.92) between experimental and predictive activity scales depicted that the generated hypothesis is able to differentiate active compounds from the inactive compounds. The least cost value for Hypo1 in Fischer’s randomization validation and high GF score in the decoy test validation also suggested its suitability to use for virtual screening of chemical databases. The validated pharmacophore was employed as 3D query for virtual screening of chemical databases. This implies that the Hypo1-retrieved hits might retain the same or improved therapeutic ability corresponding to the known inhibitors of hDHFR [43]. Molecular docking approach predicts the most compatible binding conformation of ligand molecule in the binding site of the receptor protein. Accordingly, the best poses from docking on the basis of scoring functions and essential interactions with the active site residues of hDHFR were subjected to MD simulations for stability analysis [44]. RMSD plots implied that the hits showed similar mode of interaction as the reference compound. Specifically, the average RMSD profiles (<0.25 nm) obtained for the backbone atoms of DHFR suggested that all the systems were uniform because the stability of the system can be inferred by its RMSD value less than 0.3 nm [45]. Our results showed that all the hit compounds established stable H-bonds with the active site residues of hDHFR. Intriguingly, Hit2 formed highest number of hydrogen bonds with hDHFR and NADPH. Such pattern of high number of H-bonding of inhibitors was previously observed in the development of small molecule inhibitors of DHFR of *Mycobacterium tuberculosis* [46].

In human dihydrofolate reductase, Glu30 is an essential residue in the active site and forms hydrogen bond with inhibitor(s) [18,46]. We observed that Glu30 demonstrated consistent H-bond interaction with the reference and hit compounds. It is investigated that the carboxylate oxygen of Glu30 makes H-bond interaction with the pyridopyrimidine ring of inhibitor. Likewise, N1 nitrogen and 2-NH2 of the same inhibitor form H-bonds with OE2 and OE1 of Glu30, respectively [35]. Our results also demonstrated that N21 nitrogen of Hit1 formed hydrogen bond with OE1 of Glu30 with bond distance of 1.91 Å. The interaction analysis displayed a H-bond between the oxygen atom (O27) of Hit2 and oxygen atom (OE1) of Glu30 with the bond distance of 2.03 Å. Additionally, another H-bond was observed between the O25 of Hit2 and OE2 of Glu30 with the bond distance of 1.03 Å. Our results also observed H-bond between the O11 atom of Hit3 and OE2 of Glu30. Afterward, the hydrogen bond distance in the reference and the co-crystal was observed as 2.64 Å and 2.71 Å, respectively. Therefore, we have argued that the novel hits may have a higher affinity towards DHFR than the reference compound. Furthermore, binding free energy calculations using MM/PBSA also suggested that the complexes of hDHFR with the hit compounds were more stable than the reference [47].

Therefore, the known inhibitors of hDHFR are not safe and showed severe side effects and as a result we carried out safety measures for the newly identified inhibitors. The comparative analyses of the Federal Drug Agency’s (FDA) approved drugs and the novel hits predicted that the hit compounds of hDHFR were safe and do not pose severe side effects [46,47]. The QSAR-based system generates and validates accurate and rapid assessments of chemical toxicity solely from a molecular structure. This assessment module combined with ADMET and Lipinski’s rule of five prediction modules exhibited that the hit compounds have acceptable pharmacokinetic properties. Finally, we recommend the newly identified hits of hDHFR as fundamental platforms to develop effective chemotherapeutics in cancer and rheumatoid medications.

## Figures and Tables

**Figure 1 jcm-08-00233-f001:**
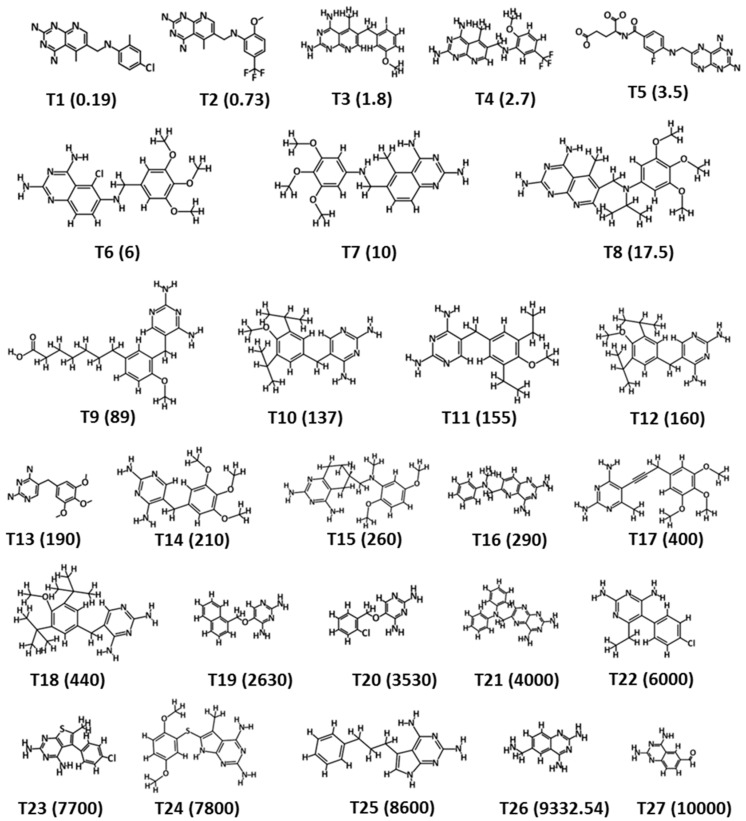
2D structure representation of the 27 chemically diverse training set compounds used for pharmacophore generation. The experimental IC_50_ values (nM/L) are shown in parentheses.

**Figure 2 jcm-08-00233-f002:**
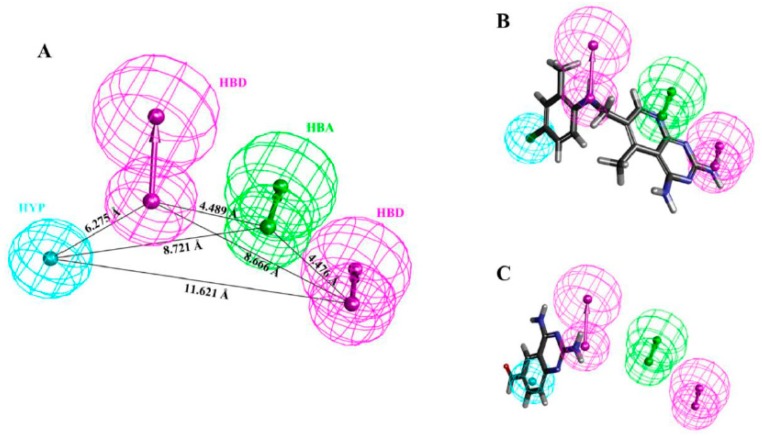
(**A**) Shows the 3D spatial arrangement and distance constraints of the chemical features of Hypo1. Hypo1 consists of one hydrogen bond acceptor (green), two hydrogen bond donor (magenta), and one hydrophobic (cyan) features. (**B**) Alignment of the reference compound (the most active compound of the training set (IC_50_ = 0.19 nM/L)) mapped all the features of the Hypo1. (**C**) Alignment of the least active compound of the training set (IC_50_ = 10,000 nM/L) mapped the hydrogen bond donor features only and missed the hydrogen bond acceptor and hydrophobic features of the Hypo1.

**Figure 3 jcm-08-00233-f003:**
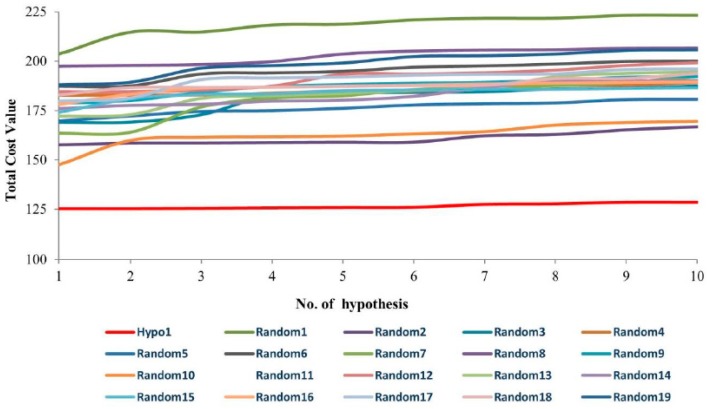
A graphical representation of the total cost values of Hypo1 and each of the 10 hypotheses generated from 19 random spreadsheets during Fischer’s randomization run. A confidence level of 95% was used. Hypo1 obtained the lowest total cost value.

**Figure 4 jcm-08-00233-f004:**
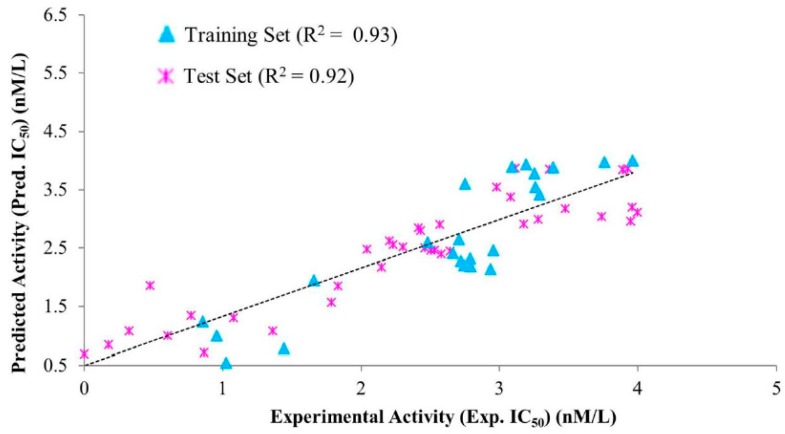
Correlations between the experimental activities and the predicted activities using Hypo1 with the training set compounds and test set compounds. Hypo1 observed strong correlation between the predicted activities of training set and test set compounds.

**Figure 5 jcm-08-00233-f005:**
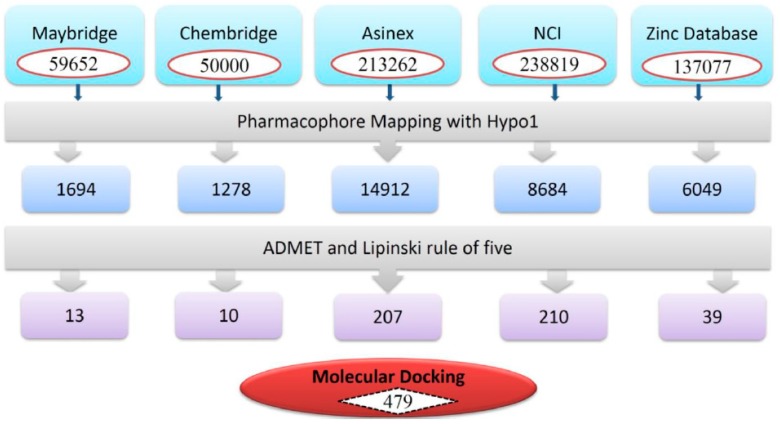
Schematic flow of the virtual screening. Hypo1 was employed as 3D query for virtual screening. ADMET assessment test and Lipiniski’s rule of five were used as filtration systems to identify drug-like compounds.

**Figure 6 jcm-08-00233-f006:**
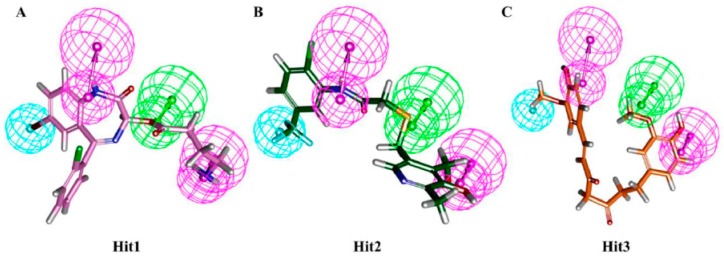
Alignment of Hypo1 and the final hit compounds. Hypo1 mapped all the features of Hit1 (**A**), Hit2 (**B**), and Hit3 (**C**). Hit compounds are represented as stick models.

**Figure 7 jcm-08-00233-f007:**
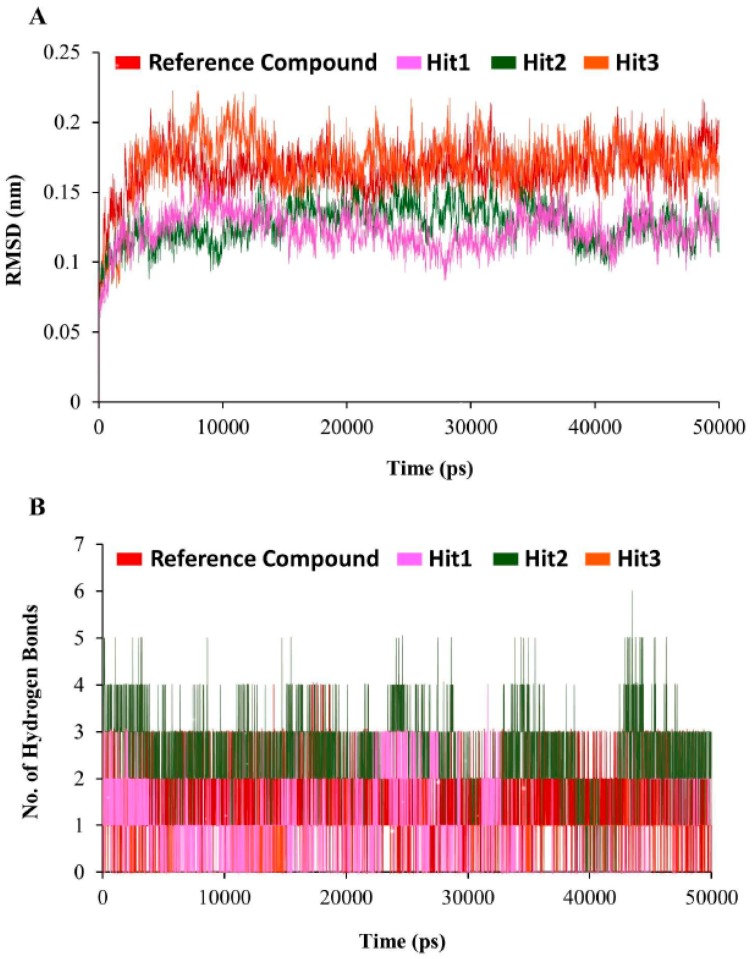
Molecular dynamics simulation analyses. (**A**) The root mean square deviation (RMSD) profile of the backbone atoms of human dihydrofolate reductase (hDHFR). Each system suggested the stable RMSD during the entire simulation run. Color scheme is depicted as hDHFR in complex with reference compound (red), Hit1 (magenta), Hit2 (green), and Hit3 (orange). (**B**) The number of intermolecular hydrogen bonds between protein and compound during the 50 ns MD simulations. The reference compound, Hit1, Hit2, and Hit3 are displayed as red, magenta, green, and orange.

**Figure 8 jcm-08-00233-f008:**
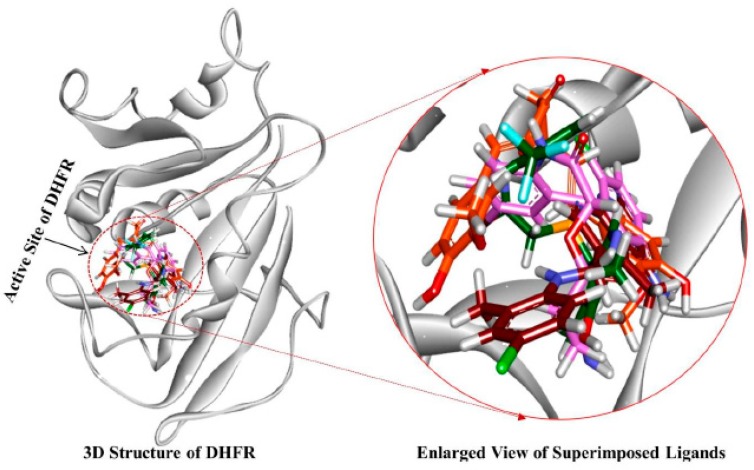
The reference compound and the three hit compounds occupied the active site of hDHFR. All the compounds in their representative structures were superimposed (left) and enlarged (right). 3D structure of hDHFR is shown in light grey color. Red, magenta, green, and orange colors represent the reference compound, Hit1, Hit2, and Hit3, respectively.

**Figure 9 jcm-08-00233-f009:**
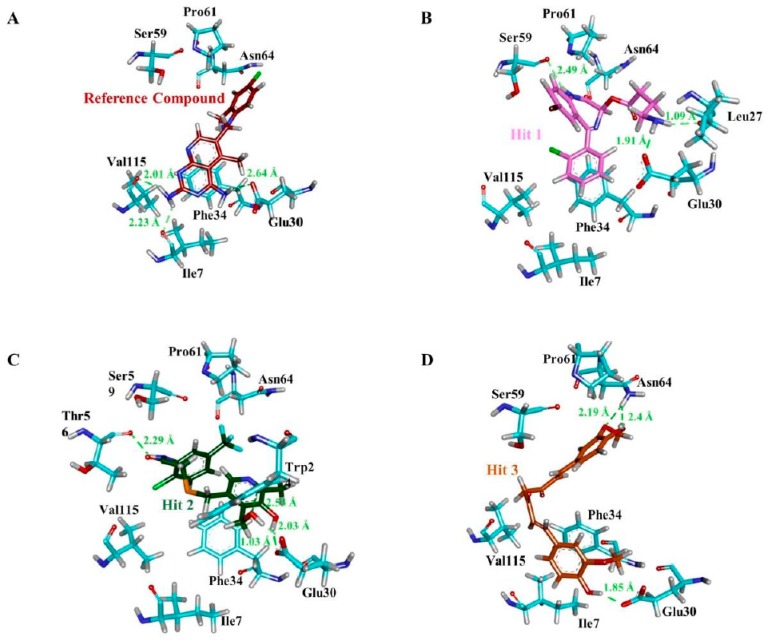
Molecular interactions analyses. The reference compound and the three hit compounds interacted essential residues in the active site of hDHFR. The reference compound (**A**), Hit1 (**B**), Hit2 (**C**), and Hit3 (**D**) are depicted as a red, magenta, green, and orange colored stick representation. The H-bond forming residues of hDHFR are displayed as cyan stick model. H-bonding and bond distance are represented as green dashed lines and measured in angstrom (Å), respectively.

**Figure 10 jcm-08-00233-f010:**
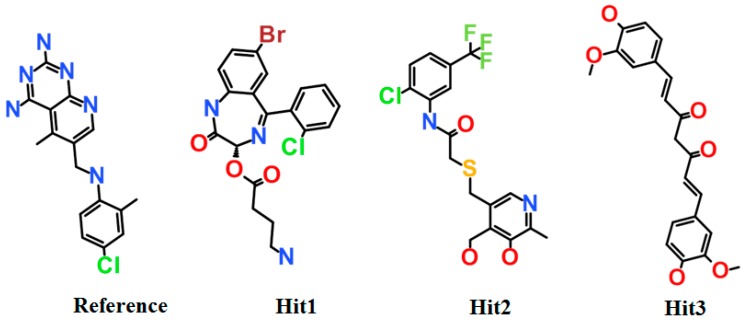
2D structures of the final hit compounds. The 2D structures of the reference compound (**A**), Hit1 (**B**), Hit2 (**C**), and Hit3 (**D**) are depicted.

**Figure 11 jcm-08-00233-f011:**
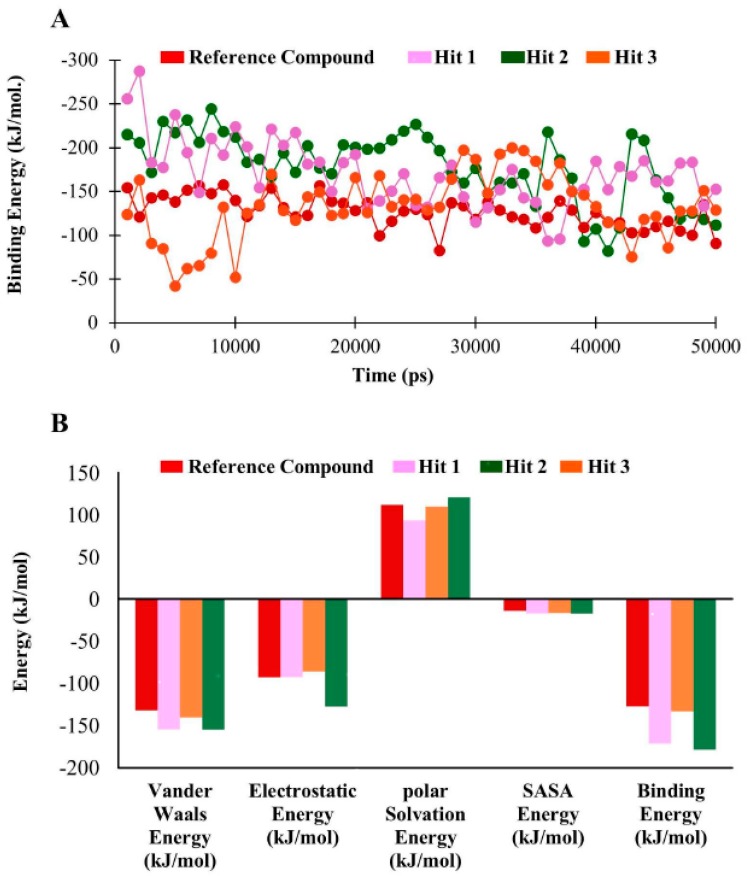
Binding free energy analyses. (**A**) Graphical representation of MM/PBSA estimated binding free energy of hDHFR in complex with reference compound, Hit1, Hit2, and Hit3 throughout the simulation time. The reference compound, Hit1, Hit2, and Hit3 are depicted as red, magenta, green, and orange, respectively. (**B**) The binding free energy decomposition analysis of the final hits in the active site of hDHFR.

**Table 1 jcm-08-00233-t001:** Statistical data of ten pharmacophore hypotheses generated by HypoGen.

Hypo. No.	Total Cost	Cost Difference ^a^	Root Means Square Deviation (RMSD) ^b^	Correlation (*R*^2^)	Max Fit	Features ^c^
Hypo1	125.276	78.308	0.99	0.94	11.15	HBA, HBD, HBD, HYP
Hypo2	125.362	78.304	1.05	0.93	9.94	HBA, HBD, HBD, HYP
Hypo3	125.509	78.157	1.05	0.93	10.04	HBA, HBD, HBD, HYP
Hypo4	125.358	77.701	1.05	0.93	10.04	HBA, HBD, HBD, HYP
Hypo5	125.867	77.799	1.06	0.93	10.21	HBA, HBD, HBD, HYP
Hypo6	125.965	77.946	1.03	0.93	10.74	HBA, HBD, HBD, HYP
Hypo7	127.533	76.133	1.13	0.92	9.97	HBA, HBD, HBD, HYP
Hypo8	125.781	75.885	1.14	0.92	9.75	HBA, HBD, HBD, HYP
Hypo9	128.568	75.098	1.15	0.92	10.24	HBA, HBD, HBD, HYP
Hypo10	128.568	75.098	1.15	0.92	10.26	HBA, HBD, HBD, HYP

^a^ Cost difference measures the difference between the null cost and the total cost. The null cost score of the ten hypotheses is 140.46 and the value of fixed cost is 86.83. The representative unit of costs is bit. ^b^ RMSD measures the deviation of the log of estimated activities from the log of experimental activities and is normalized by the log of uncertainties. ^c^ Features, HBD, HBA, and HYP represent hydrogen bond donor, hydrogen bond acceptor, and hydrophobic, respectively.

**Table 2 jcm-08-00233-t002:** Experimental and predicted activity of training set compounds based on Hypo1.

Compound No.	Fit Value	Experimental IC_50_ nM/L	Predicted IC_50_ nM/L	^a^ Error	^b^ Exp. Scale	^b^ Pred. Scale
1	9.63	0.19	0.21	1.10	+++	+++
2	7.42	0.73	7.55	10.34	+++	+++
3	6.87	1.8	27.09	15.05	+++	+++
4	8.24	2.7	1.15	−2.36	+++	+++
5	7.33	3.5	9.40	2.69	+++	+++
6	7.66	6	4.41	−1.36	+++	+++
7	7.32	10	9.68	−1.03	+++	+++
8	7.36	17.5	8.69	−2.01	+++	+++
9	6.26	89	110.31	1.24	+++	++
10	5.74	137	364.60	2.66	++	++
11	5.75	155	357.03	2.30	++	++
12	5.77	160	339.73	2.12	++	++
13	5.52	190	606.05	3.19	++	+
14	5.78	210	331.91	1.58	++	++
15	5.62	260	482.36	1.86	++	++
16	5.38	290	827.48	2.85	++	+
17	5.75	400	352.12	−1.14	++	++
18	5.80	440	315.88	−1.39	++	++
19	5.11	2630	1566.48	−1.68	+	+
20	4.84	3530	2915.70	−1.21	+	+
21	5.18	4000	1327.90	−3.01	+	+
22	4.78	6000	3296.55	−1.82	+	+
23	5.01	7700	1946.96	−3.95	+	+
24	5.06	7800	1749.88	−4.46	+	+
25	5.53	8600	597.43	−14.40	+	+
26	4.52	9332	6062.97	−1.54	+	+
27	4.34	10,000	9094.55	−1.10	+	+

^a^ Error: ratio of the predicted activity (Predicted IC_50_) to the experimental activity (Experimental IC_50_) or its negative inverse if the ratio is <1. ^b^ Activity scale: IC_50_ < 100 nM/L = +++ (active), 100 nM/L < IC_50_ < 500 nM/L = ++ (moderate active), IC_50_ ≥ 500 nM/L = + (inactive).

**Table 3 jcm-08-00233-t003:** Decoy set validation. The decoy set validation of Hypo1 obtained highest goodness of fit (GF) score (0.86) and suggested its suitability for virtual screening.

Parameters	Values
Total no. of molecules in database (D)	75
Total no. of actives in database (A)	8
Total no. of hit molecules from the database (Ht)	8
Total no. of active molecules in hit list (Ha)	7
Percentage Yield of actives ((Ha/Ht) × 100)	87.5%
Percentage Ratio of actives ((Ha/A) × 100)	88%
Enrichment Factor (EF = (Ha/Ht)/(A/D))	8.23
False negatives (A − Ha)	1
False positive (Ht − Ha)	1
Goodness of fit score (GF)	0.86

**Table 4 jcm-08-00233-t004:** Comparison of Goldscore and Chemscore of dihydrofolate reductase (DHFR) in complex with the reference, Hit1, Hit2, and Hit3.

System	Goldfitness Score	Chemscore	Estimated IC_50_ (nM/L)
DHFR + ^a^ Reference	44.67	−23.35	0.21
DHFR + Hit1	70.05	−34.51	0.12
DHFR + Hit2	58.95	−33.66	0.043
DHFR + Hit3	57.31	−37.27	0.17

^a^ Reference: the most active compound in the training set.

**Table 5 jcm-08-00233-t005:** The specifications of four systems used for molecular dynamics simulations.

System	No. of TIP3P Water Molecules	No. of Na^+^ Ions	System Size (nm)
DHFR+NADPH + ^a^ Reference	11,306	4	7.193 × 7.193 × 7.193
DHFR+NADPH + Hit1	11,306	4	7.193 × 7.193 × 7.193
DHFR+NADPH + Hit2	11,306	4	7.193 × 7.193 × 7.193
DHFR+NADPH + Hit3	11,306	4	7.193 × 7.193 × 7.193

^a^ Reference: the most active compound in the training set.

**Table 6 jcm-08-00233-t006:** Molecular interactions between the ligands (the reference and hit compounds) and the active site residues of hDHFR.

Compound	Hydrogen Bond (<3 Å)	Van der Waals Interactions and Carbon Hydrogen Bond	π-Interaction
^a^ Reference	Ile7, Glu30, Val115	Val8, Gly31, Asn64, Gly116, Tyr121, NADPH	Ile7, Phe34, Ile60, Pro61, Leu67, Val115
Hit1	Leu27, Glu30, Ser59	Val8, Ile16, Gly17, Asp21, Asp21, Pro26, Gly31, Gln35, Thr136, Pro61, Ile60	Ala9, Leu22, Pro23, Phe34, Val115, NADPH
Hit2	Trp24, Glu30 (2), Thr56, NADPH	Ala9, Gly31, Gln35, Ser59, Ile60, Pro61	Phe34, Met52, Ile60, Leu67, Val115
Hit3	Glu30, Asn64 (2)	Val8, Leu22, Gly31, Gln35, Met52, Thr56, Pro61, Leu67, Val115, Tyr121, Thr136, NADPH	Ile7, Ala9, Ile60, Phe34

^a^ Reference: the most active compound in the training set.

**Table 7 jcm-08-00233-t007:** Decomposition of binding free energy.

Compounds	Van der Waals Energy (kJ/mol)	Electrostatic Energy (kJ/mol)	Polar Solvation Energy (kJ/mol)	SASA ^b^ Energy (kJ/mol)	Binding Energy (kJ/mol)
^a^ Reference	−131.78	−92.84	111.73	−13.9	−127.05
Hit1	−154.58	−92.63	93.27	−17.218	−171.12
Hit2	−154.75	−127.45	120.79	−17.31	−178.47
Hit3	−140.36	−86.23	109.75	−16.47	−133.16

^a^ Reference: the most active compound in the training set. SASA ^b^: Solvent accessible surface area.

**Table 8 jcm-08-00233-t008:** Comparison of absorption, distribution, metabolism, excretion, and toxicity (ADMET), Lipinski’s rule of five, and toxicity prediction by komputer assisted technology (TOPKAT) predictions for Federal Drug Agency (FDA) approved drugs and hit compounds.

Name	ADMET Analysis			Lipinski’s Rule of Five				TOPKAT Analysis		
	Solubility	BBB ^a^	Hepato-Toxicity	Absorption	HBA ^b^	HBD ^c^	M.W (Da) ^d^	Rat (Carcinogenicity)		Skin Irritancy
								Female	Male	
MTX	2	4	TRUE	3	12	5	454.447	Single	Non	Non
Pralatexet	2	4	TRUE	3	11	5	477.481	Non	Non	Non
Pemetrexed	3	4	TRUE	3	6	6	427.417	Single	Non	Non
Hit1	3	3	FALSE	0	6	3	450.714	Non	Non	Non
Hit2	3	3	FALSE	0	5	3	420.833	Non	Non	Non
Hit3	3	3	FALSE	0	6	2	368.385	Non	Non	Non

^a^ BBB: blood brain barrier level. ^b^ HBA: hydrogen bond acceptor. ^c^ HBD: hydrogen bond donor. ^d^ M.W (Da): molecular wright in Dalton.

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
