# Peer review of "In Silico Study Probes Potential Inhibitors of Human Dihydrofolate Reductase for Cancer Therapeutics"

_jcm, 2019, doi:10.3390/jcm8020233_

Reviewer 1 Report

Manuscript jcm-428185, entitled "In Silico Study Probes Potential Inhibitors of Human Dihydrofolate Reductase for Cancer Therapeutics" by Rana, R.M. et al. regards the in silico identification of three molecules that could bind and inhibit  DHFR with a safer biological profile with respect to the reference compound MTX. In my opinion this is a good manuscript, but I suggest the following improvements:

Introduction

-pag. 1 line 41: substitute "which lead" with "leading";

pag.2 line 47: which is the "molecule" "bound with an almost planar arrangement"? Please, explicit it

- The description of the DHFR catalytic mechanism  (lines 57-60) is not clear;

-Phrase from line 61 to line 63 is a repetition and should be deleted

Results

-pag.5 line 185: the word "inhibitor" should be "inhibitors"

-pag.6 Figure 1 resolution is very low and caption (line 194) wrongly reports 24 compounds instead of 27.

-pag.7 line 208: insert "for" after except

-pag 7 lines 208-209: change "where they" with "which". Moreover, say which the underestimated compounds are, as, from table 2, the underestimated compounds seem to be three (cpds 9,13 and 16)

-pag 12 line 307: the phrase "A total of four MD simulation systems;" is not complete

-pag16 line 363. Include a separate figure of the 2D chemical structures of HIT1, HIT 2 and HIT 3 as from figure 9 they are not clear

Finally I suggest to include also the predicted IC50 for the selected HIT1, HIT 2 and HIT 3

Author Response

Response to reviewers

We certainly appreciate all the reviewers for being positive about our manuscript and to provide highly constructive suggestions to improve our manuscript.

Reviewer comments:

Reviewer 1

Reviewer 1 marked that the author “must improve” the background.

We appreciate Reviewer 1 to point out the shortcomings of our manuscript and let us a chance to improve it. We have thoroughly updated the paper with special focus on introduction/background section. Please refer to the refined version.

Reviewer 1 marked that research design, methodology and results are adequetly described and clearly presented.

We appreciate reviewer 1 for being positive and acceptance of our research out.

Reviewer 1 marked that conclusions support the results.

We highly appreciate reviewer 1 for his complement.

-pag. 1 line 41: substitute "which lead" with "leading";

“Leading” is inserted

pag.2 line 47: which is the "molecule" "bound with an almost planar

arrangement"? Please, explicit it

This has modified, please refer to introduction.

- The description of the DHFR catalytic mechanism (lines 57-60) is not clear;

Modified in introduction, please refer to introduction.

-Phrase from line 61 to line 63 is a repetition and should be deleted

This has also been updated.

Results

-pag.5 line 185: the word "inhibitor" should be "inhibitors"

Replaced

-pag.6 Figure 1 resolution is very low and caption (line 194) wrongly reports

24 compounds instead of 27.

Has been updated

-pag.7 line 208: insert "for" after except

Has been inserted

-pag 7 lines 208-209: change "where they" with "which". Moreover, say which

the underestimated compounds are, as, from table 2, the underestimated

compounds seem to be three (cpds 9,13 and 16)

“where they” is replaced by “which”.

Yes, they are three and we have mentioned that one active and two moderate active, so total are three

-pag 12 line 307: the phrase "A total of four MD simulation systems;" is not

complete

this sentence has been modified. Please refer to line no. 363 of the refined manuscript.

-pag16 line 363. Include a separate figure of the 2D chemical structures of

HIT1, HIT 2 and HIT 3 as from figure 9 they are not clear

The 2D structures were already been included in the supplementary information. Please refer to Figure S3.

Finally I suggest to include also the predicted IC50 for the selected HIT1, HIT

2 and HIT 3

We tried to find out some tool which can predict the IC50 value of a compound but could not find. Also the prediction of IC50 from protein-ligand interaction is not encouraging in literature survey.

Comments in general

1.      The entire manuscript was revisited and checked for English errors by two colleagues who are good in English speaking.

2.      Changes were made thoroughly wherever they were needed.

3.      Figure 1 was updated because of the low resolution of its previous figure.

      4.      References were rearranged and formatted by Mendeley reference manager.

Reviewer 2 Report

Rana R.M & al. focused their study on DHFR (E.C. 1.5.1.3) as potential target in cancer therapy. However, the manuscript is not easy to follow, the title and the overall workflow is quite ambiguous.

1.       Introduction. The implication of DHFR in different diseases could be arranged more logical  – e.g. the rheumatoid arthritis is mentioned many times – lines 46, 62, 67

-          Line 47-60 In between paragraphs concerns with description of DHFR links with pathological conditions, there are details of DHFR conformation and enzyme interaction. For a reader non-familiar with DHFR these details are not very informative –e.g – line 54 – “the most relevant molecular interactions” – relevant in what sense?

-          The references are not always reflect the ideas from the main text

2.       Materials and Methods. The study seems to follow the same workflow designed in the authors previous work (PLOS ONE | DOI:10.1371/journal.pone.0147190 January); In my opinion the study workflow is not so clear explained in the present manuscript.

-          It is not clear how the dataset was designed. Line 81 –“literature mining” (some references or some public compounds  repository should be added)

-          Lines 83-85 – from where were obtained the “experimentally determined IC50 values” – the input data for model construction (references / where these compounds originate?)

-          Line 84 - Define “active,  moderate, inactive compounds” in the first mentioning in the text

-          line135 – PDB ID must be written in the first mentioning in the text, also stressed why the authors selected 1BOZ from the protein data bank?

-          lines 142, 151 explanation of the abbreviations in the first mentioning in the text

3.       Results

-          Line 185-186 – since it is not explained how the 27 DHFR inhibitors are selected from the original 67 original dataset it is difficult to follow the main text and the Figure 1.  

-          Table 2 – the data are not enough explained in the main text

-         line 447-449 Since the authors identified non-toxic inhibitors of DHFR, why not experimental validate their promising results in the cancer and rheumatoid treatment?  

4.       References – are it written according to Molecules requirements?  

5.       Typing errors.

-          Many dashes where no needed – see lines 51-53 e.g. Arg-28 should be Arg28

-          Lines 41-43 phrase “ Since, DHFR is a rate ….. , therefore DHFR … anticancer drugs and antibiotics” sounds weird to me.

-          Line 69 - ?complexities

Author Response

Response to reviewers

We certainly appreciate all the reviewers for being positive about our manuscript and to provide highly constructive suggestions to improve our manuscript.

Reviewer comments:

Reviewer 2

We appreciate reviewer 2 to evaluate our manuscript and showed interest in our work. We’re thankful to reviewer 2 for his constructive suggestions and/or comments to make our manuscript more clear and reader-friendly.

Reviewer 2 marked that the author “must improve” the background.

We appreciate Reviewer 1 to point out the shortcomings of our manuscript and let us a chance to improve it. We have thoroughly updated the paper with special focus on introduction/background section. Please refer to the refined version.

Reviewer 2 marked that research designing can be improved.

We have explained the goal of our research. Please refer to the refined version of the manuscript.

Reviewer 2 suggested improving methods and results.

We have updated both the methods and results sections. Please refer to the correspondent sections in the refined manuscript.

1.      Introduction. The implication of DHFR in different diseases could be arranged more logical – e.g. the rheumatoid arthritis is mentioned many times – lines 46, 62, 67

The introduction is improved with special focus on the problem, and its solution.

- Line 47-60 In between paragraphs concerns with description of DHFR links with pathological conditions, there are details of DHFR conformation and enzyme interaction. For a reader non-familiar with DHFR these details are not very informative –e.g – line 54 – “the most relevant molecular interactions” – relevant in what sense?

The comment has been addressed. Please refer to the introduction part.

- The references are not always reflect the ideas from the main text

The references were revisited and the main text was improved.

2.      Materials and Methods. The study seems to follow the same workflow designed in the authors previous work (PLOS ONE | DOI:10.1371/journal.pone.0147190 January); In my opinion the study workflow is not so clear explained in the present manuscript.

- It is not clear how the dataset was designed. Line 81

–“literature mining” (some references or some public compounds repository should be added)

The link of Binding database (https://www.bindingdb.org/bind/index.jsp) has been added.

      - Lines 83-85 – from where were obtained the “experimentally determined IC50 values” – the input data for model construction (references / where these compounds originate?)

These compounds were collected from binding database and the link is inserted in the main text. Please refer to the Material and Methods section of the manuscript.

- Line 84 - Define “active, moderate, inactive compounds” in

the first mentioning in the text

The classification was made on the basis of IC50 value. The criteria are added to differentiate between these classes. Please refer to Material and Methods section.

- line135 – PDB ID must be written in the first mentioning in the text, also stressed why the authors selected 1BOZ from the protein data bank?

PDB ID of DHFR has written in the first mentioning in the text. Please refer to line no. 155 of the refined manuscript.

The PDB ID has been added, please refer the Material and Methods section. Since, we were interested to find out the binding affinity and mechanism of the newly identified inhibitors with DHFR, that’s why we used DHFR structure from PDB. As per criteria, there is no such hard and fast roles to select PDB structure for docking but still we are taking care of certain parameters like the structure should be complete, there should be an already bound inhibitor to compare our findings, the structure should not have deleterious mutations, the structure should have high resolution etc.

- lines 142, 151 explanation of the abbreviations in the first mentioning in the text

For GOLD, the abbreviation is already given in line 152 of the refined version of the manuscript. For other acronyms, the explanations are inserted on their first appearance. Please refer to the main text.

3. Results

- Line 185-186 – since it is not explained how the 27 DHFR

inhibitors are selected from the original 67 original dataset it is

difficult to follow the main text and the Figure 1.

This section has been modified in the main text. Please refer to “Generation of Pharmacophore Model” section of the main text.

- Table 2 – the data are not enough explained in the main text

The correspondent section has modified. Please refer to line no. 255-261 the main text.

- line 447-449 Since the authors identified non-toxic inhibitors of DHFR, why not experimental validate their promising results in the cancer and rheumatoid treatment?

We’re sorry to say that we couldn’t perform the experimental validation yet. Even though we have plan to find an experimental group working on DHFR inhibition and could validate the activity of these compounds in future studies.

4. References – are it written according to Molecules requirements?

Yes, the references were formatted by Mendeley – Reference management software.

5. Typing errors.

- Many dashes where no needed – see lines 51-53 e.g. Arg-28 should be Arg28

Yes, We fixed such errors.

- Lines 41-43 phrase “ Since, DHFR is a rate ….. , therefore

DHFR … anticancer drugs and antibiotics” sounds weird to me.

This section has been improved. Please refer to results section.

- Line 69 - ?complexities

This terminology has replaced by consequences. Please refer to the final paragraph of the introduction section.

Comments in general

1.      The entire manuscript was revisited and checked for English errors by two colleagues who are good in English speaking.

2.      Changes were made thoroughly wherever they were needed.

3.      Figure 1 was updated because of the low resolution of its previous figure.

      4.      References were rearranged and formatted by Mendeley reference manager.

Reviewer 3 Report

The manuscript by Park et al. aims to predict effective and safe inhibitors of human DHFR for cancer and rheumatoid arthritis chemotherapeutics. I recommend its publication after minor revision. My comments are as follows:

1.     What were the authors’ criteria for selecting molecules for their training set and test set? Where did the used experimental activity data come from?

2.     References should be cited to all software/programs used for calculations.

3.     The choice of PDB protein structure 1BOZ used for docking studies should be explained.

4.     The list of references should be checked carefully for incomplete citations. Some of the parameters are missing (e.g. the source of the article and/or date of publication in refs 1, 2, 11-14, 33).

Author Response

Response to reviewers

We certainly appreciate all the reviewers for being positive about our manuscript and to provide highly constructive suggestions to improve our manuscript.

Reviewer comments:

Reviewer 3

We highly appreciate reviewer 3 for his effort to read our manuscript and gave valuable suggestions which may contribute more to our manuscript.

Reviewer 3 marked that the author can improve the background.

We have improved the introduction section. Please refer to the refined manuscript.

Reviewer 3 marked that research is appropriately designed, results are clearly presented and conclusions are supported by the results.

We’re thankful to reviewer 3 for his complement.

Reviewer 3 has marked that the author can improve the methods.

We made significant changes to methodology. Please refer to the refined version.

Comments and Suggestions for Authors:

The manuscript by Park et al. aims to predict effective and safe inhibitors of

human DHFR for cancer and rheumatoid arthritis chemotherapeutics. I

recommend its publication after minor revision. My comments are as follows:

1. What were the authors’ criteria for selecting molecules for their training

set and test set? Where did the used experimental activity data come from?

The compounds of the training and test sets were taken from literature survey. These compounds were already tested against the DHFR while using the same assays protocols. We chose a wide range for the selection dataset compounds based on: 1) should be tested as DHFR inhibitor, 2) should have structural diversity, 3) and different IC50 values.

The experimental activity data of the selected compounds were mined from their respective published reports.

2. References should be cited to all software/programs used for

calculations.

Yes, references were added to all the software/programs. Please refer to the main text.

3. The choice of PDB protein structure 1BOZ used for docking studies

should be explained.

Yes, this section is updated in the main text. Please refer to line no. 153-157.

 4. The list of references should be checked carefully for incomplete

citations. Some of the parameters are missing (e.g. the source of the article

and/or date of publication in refs 1, 2, 11-14, 33).

Yes, the entire list was revisited and fixed the missing parts.

Comments in general

1.      The entire manuscript was revisited and checked for English errors by two colleagues who are good in English speaking.

2.      Changes were made thoroughly wherever they were needed.

3.      Figure 1 was updated because of the low resolution of its previous figure.

4.      References were rearranged and formatted by Mendeley reference manager.

Round  2

Reviewer 1 Report

Although Authors have ameliorated the manuscript  according to my suggestion the two following concerns must be addressed:

1) It is essential to readily examine HIT1, HIT2 and HIT3 chemical structures when reading the work, therefore they should be reported in the article, not in supplementary information. 

2)Based on the reported Test Set Validation, explain why Hypo1 was not used to predict/verify HITs1-3 IC50 values, as for the test set compounds.

Author Response

Response to reviewer 1

We highly appreciate reviewer 1 for being positive about our manuscript and to provide highly constructive suggestions to improve our manuscript.

Comments for authors

1)      It is essential to readily examine HIT1, HIT2 and HIT3 chemical structures when reading the work, therefore they should be reported in the article, not in supplementary information.

We highly appreciate reviewer 1 for his suggestion. We have moved supplementary figure 3 to the main figures and number as figure 10. The subsequent figures number is updated accordingly. Please refer to line number 430-439 of the revised manuscript.

2)      Based on the reported test set validation, explain why Hypo1 was not used to predict/verify HITs1-3 IC50 values, as for the test set compounds.

We appreciate reviewer 1 for this comment to repeat it more clearly. In the first attempt, we responded in different meaning to this comment. Yes, we have calculated the estimated IC50 values of the newly identified hits by Hypo1 and the values are added to the main text. Please refer to table 4 and line number of 355-357 of the revised manuscript.

Reviewer 2 Report

If the J. Clin. Med. accept theoretical studies, this paper can be published.

Author Response

Reviewer 2

We're grateful to reviewer 2 for his recommendation of our manuscript for possible publication in Journal of Clinical Medicine.